# Low-Molecular-Weight Heparin Reduces Ventilation-Induced Lung Injury through Hypoxia Inducible Factor-1α in a Murine Endotoxemia Model

**DOI:** 10.3390/ijms21093097

**Published:** 2020-04-28

**Authors:** Li-Fu Li, Yung-Yang Liu, Shih-Wei Lin, Chih-Hao Chang, Ning-Hung Chen, Chen-Yiu Hung, Chung-Shu Lee

**Affiliations:** 1Department of Internal Medicine, Division of Pulmonary and Critical Care Medicine, Chang Gung Memorial Hospital, Taoyuan 333, Taiwan; 2Department of Internal Medicine, Chang Gung University, Taoyuan 333, Taiwan; 3Department of Respiratory Therapy, Chang Gung Memorial Hospital, Taoyuan 333, Taiwan; 4Chest Department, Taipei Veterans General Hospital, Taipei 112, Taiwan; 5Institutes of Clinical Medicine, School of Medicine, National Yang-Ming University, Taipei 112, Taiwan

**Keywords:** ventilator-induced lung injury, endotoxemia, low-molecular-weight heparin, hypoxia-inducible factor

## Abstract

Patients with sepsis frequently require mechanical ventilation (MV) to survive. However, MV has been shown to induce the production of proinflammatory cytokines, causing ventilator-induced lung injury (VILI). It has been demonstrated that hypoxia-inducible factor (HIF)-1α plays a crucial role in inducing both apoptotic and inflammatory processes. Low-molecular-weight heparin (LMWH) has been shown to have anti-inflammatory activities. However, the effects of HIF-1α and LMWH on sepsis-related acute lung injury (ALI) have not been fully delineated. We hypothesized that LMWH would reduce lung injury, production of free radicals and epithelial apoptosis through the HIF-1α pathway. Male C57BL/6 mice were exposed to 6-mL/kg or 30-mL/kg MV for 5 h. Enoxaparin, 4 mg/kg, was administered subcutaneously 30 min before MV. We observed that MV with endotoxemia induced microvascular permeability; interleukin-6, tumor necrosis factor-α, macrophage inflammatory protein-2 and vascular endothelial growth factor protein production; neutrophil infiltration; oxidative loads; HIF-1α mRNA activation; HIF-1α expression; bronchial epithelial apoptosis; and decreased respiratory function in mice (*p* < 0.05). Endotoxin-induced augmentation of VILI and epithelial apoptosis were reduced in the HIF-1α-deficient mice and in the wild-type mice following enoxaparin administration (*p* < 0.05). Our data suggest that enoxaparin reduces endotoxin-augmented MV-induced ALI, partially by inhibiting the HIF-1α pathway.

## 1. Introduction

Sepsis is the leading cause of admission to intensive care units (ICUs) and results in a hospital mortality rate as high as 30%. During the course of sepsis, infectious components, such as lipopolysaccharide (LPS), may induce an inflammatory cascade causing the release of inflammatory cytokines interleukin (IL)-6, macrophage inflammatory protein-2 (MIP-2), tumor necrosis factor (TNF)-α and vascular endothelial growth factor (VEGF), which can engender microvascular leakage, lung edema, epithelial and endothelial injury and hypoxemia. This can lead to multiple organ failure [1,2,3]. Mechanical ventilation (MV) is often used to maintain the life of patients with sepsis-related acute lung injury (ALI). However, prolonged MV may increase the risk of pathologic overdistention in the lungs, increase the production of inflammatory cytokines, recruit inflammatory cells, and eventually induce a type of ALI, termed ventilator-induced lung injury (VILI) [4]. Accumulating evidence has revealed that mechanically ventilating patients is associated with lung inflammation, increased oxidative stress and lung tissue hypoxia, which may contribute to difficulty in ventilator weaning and complications such as ventilator-associated pneumonia, sepsis, and increased mortality [1,2,3,4]. However, the mechanisms regulating the interactions between sepsis, MV, and these inflammatory reactions remain unclear.

Heparin has a therapeutic role in the management of sepsis because of its anti-inflammatory and anticoagulant actions [5,6]. Heparin can be categorized into unfractionated or low-molecular-weight heparin (LMWH) [7]. LMWH is a promising alternative anticoagulant owing to its superior antithrombotic effect, improved bioavailability and efficacy and decreased risk of bleeding [7]. Enoxaparin, one type of LMWH, is easier and safer for clinical use than unfractionated heparin. In our previous study of an animal VILI model, LMWH significantly decreased microvascular permeability, neutrophil influx, pulmonary plasminogen activator inhibitor-1 (PAI-1) gene expression, generation of active PAI-1 and lung injury scores due to its anti-inflammatory properties and through counteracting dysregulated coagulation and fibrinolysis [8]. Nevertheless, the beneficial effects and molecular mechanisms of LMWH on sepsis-induced lung injury have not been investigated in a preclinical study on mechanically ventilated animals.

Hypoxia-inducible factor (HIF)-1 is a heterodimeric protein, consisting of an oxygen-sensitive HIF-1α subunit with a special oxygen-dependent degradation domain and a constitutively expressed HIF-1β subunit that is critical for modulating oxygen homeostasis [9]. HIF-1α is a transcription factor composed of oxygen-regulated subunits and is expressed in all mammalian cell types [9]. Under normoxic conditions, HIF-1α is regulated in an oxygen-sensitive manner; it is hydroxylated by prolyl hydroxylases (PHDs) and recognized by the von Hippel–Lindau ubiquitin ligase, which results in polyubiquitination and proteasomal degradation [10]. Under hypoxic conditions, HIF-1α stabilizes, translocates into the nucleus, and determines the action of HIF-1α. HIF-1α takes part in the transcriptional activation of hypoxia-responsive genes by binding to the genes’ hypoxia-response element in the enhancer or promoter region [10]. Moreover, previous studies of sepsis and ALI have demonstrated that HIF-1α expression may induce acute lung epithelial damage and the production of numerous proinflammatory cytokines such as IL-6, MIP-2 and TNF-α via the nuclear factor-κB (NF-κB) pathway [11,12,13,14,15]. One important mechanism is the transcriptional regulation of HIF-1α by the redox-sensitive transcription factor NF-κB, which binds at a distinct element in the proximal promoter of the HIF-1α gene [16]. Specifically, TNF-α and MIP-2 are primary cytokines known to trigger a variety of clinical manifestations of LPS-induced sepsis, whereas IL-6 levels correlate to a poor outcome and multiple organ failure in septic patients [11]. VEGF is also crucial in the regulation of endothelial injury and vascular permeability related to ALI [17]. In addition to its anti-coagulation ability, LWMH was demonstrated to exert anti-inflammatory effects through inhibiting NF-κB pathway in LPS-treated human alveolar macrophages [18,19]. Although the effects of HIF-1α on VILI and sepsis in hypoxemia and sepsis models are already known, the beneficial effects of LMWH in the treatment of sepsis-related lung inflammation through HIF-1α pathway have not been investigated. Using our VILI model in mice after LPS challenge, our objectives were to investigate the effects of LMWH on (1) HIF-1α expression associated with the development of lung injury during MV; (2) oxidative stress, inflammatory cytokine production and endotoxin-augmented lung injury; (3) HIF-1α signaling in the VILI with sepsis; and (4) HIF-1α signaling in apoptosis of bronchial epithelia with sepsis. We hypothesized that mechanical stretch with or without LPS would augment lung injury, the production of free radicals and epithelial apoptosis, and LMWH would attenuate VILI in endotoxemic mice via the HIF-1α pathway.

## 2. Results

### 2.1. The Effects of Endotoxin Stimulation on VILI Are Partially Inversed by Enoxaparin

Mice were administered either high-tidal-volume (V_T_ = 30 mL/kg) or low-tidal-volume (V_T_ = 6 mL/kg) MV with room air for 5 h to induce VILI. The physiological conditions at the beginning and end of MV are presented in Appendix A. Stable hemodynamic status was maintained by monitoring the mean arterial pressure of the mice. Histological examinations (Figure 1A,B: alveolar congestion, hemorrhage, neutrophil infiltration and thickness of alveolar wall), gross pathological results (Figure 1B), quantitative analysis of lung histopathology (Figure 1C), gas exchange [partial pressure of oxygen (PaO_2_)/fraction of inspired oxygen (FiO_2_)] (Figure 1D) and respiratory function enhanced pause (Penh) (Figure 1E) indicated that the animal lungs injured by high-tidal-volume MV with endotoxemia displayed a hemorrhaging pattern, severe congestion, hypoxemia and an increase of lung resistance (Figure 1). Moreover, the injurious effects of hyperinflation-induced alterations of microvascular permeability and lung edema in VILI were identified by measuring lung wet-to-dry weight ratio, bronchoalveolar lavage (BAL) fluid total protein and inflammatory cytokines (Figure 2). Increased levels of wet-to-dry weight ratio; BAL fluid total protein; and IL-6, TNF-α, MIP-2 and VEGF protein production were observed in mice with endotoxemia treated with high-tidal-volume compared with the other MV treatment groups and the nonventilated control mice (Figure 2). The increases in lung inflammation and deterioration of gas exchange and respiratory function in the high-tidal-volume MV and endotoxemia mice were substantially suppressed after the administration of enoxaparin (Figure 1 and Figure 2). The effects of enoxaparin treatment in low-tidal-volume ventilated animals were demonstrated in Appendix A.

### 2.2. The Effects of Endotoxin-Stimulated MV-Induced Neutrophil Sequestration, Oxygen Radicals and HIF-1α mRNA Activation Are Partially Inhibited by Enoxaparin

The infiltration of neutrophils, a potential source of reactive oxygen species (ROS), in the marginated alveoli, vasculature and lung parenchyma; oxidative load; and antioxidant capacity were assessed to determine the lung inflammation and oxidative stress in deteriorating VILI (Figure 3A–C). Increased levels of myeloperoxidase (MPO) and protein carbonyl groups and reduced production of sodium dismutase (SOD) were evident in mice with endotoxemia treated with V_T_ 30 mL/kg compared with the other MV treatment groups and the nonventilated control mice (Figure 3A–C). To determine whether the increased neutrophil influx in mice receiving high-tidal-volume MV was associated with upregulation of chemotactic factors for neutrophils, real-time polymerase chain reaction (PCR) was performed to identify the effects of MV on endotoxin-related HIF-1α mRNA expression (Figure 3D). The upregulation of HIF-1α mRNA expression was larger in mice with endotoxemia treated with high-tidal-volume MV compared with the other MV treatment groups and the nonventilated control mice (Figure 3D). However, a reversal of these findings was observed after the administration of enoxaparin (Figure 3). Mice with endotoxemia and that received MV displayed more extensive lung damage than those without endotoxemia treated with high-tidal-volume MV (Figure 1, Figure 2 and Figure 3), which indicated the synergistic effects of LPS treatment.

### 2.3. The Effects of Endotoxin-Stimulated MV-Induced HIF-1α Expression Are Partially Reduced by Enoxaparin

Because HIF-1α upregulation has been demonstrated to modulate stretch-induced ALI in previous studies, we measured HIF-1α expression to investigate the role of the HIF-1α pathway in VILI (Figure 4 and Appendix A) [11,13,15]. Western blot analyses revealed increased HIF-1α expression in mice with endotoxemia treated with high-tidal-volume MV compared with the other MV treatment groups and the nonventilated control mice. Moreover, the increase of HIF-1α expression in mice with endotoxemia treated with high-tidal-volume MV was substantially reduced by inhibition with enoxaparin (Figure 4A). We used immunohistochemistry to further confirm the effects of HIF-1α activation in endotoxin-induced VILI (Figure 4B,C). Positive results of immunohistochemical staining for HIF-1α were substantially increased in the airway epithelial cells of the mice with endotoxemia treated with high-tidal-volume MV compared with the other MV treatment groups and the nonventilated control mice (Figure 4B,C). Consistent with the western blot results, the increases in HIF-1α expression after MV were substantially mitigated by inhibition with enoxaparin (Figure 4B,C).

### 2.4. The Effects of Endotoxin-Stimulated MV-Induced Lung Injuries, Hypoxemia and Impaired Respiratory Function Are Partially Inhibited in HIF-1α Deficient Mice

HIF-1α-deficient mice were employed to determine the role of HIF-1α in stretch-induced lung damage by examining whether the improvements in lung injuries following the administration of endotoxin was induced through HIF-1α expression. The effects of MV on different parameters, including changes to microvascular permeability and lung edema, inflammatory cytokines generation, neutrophil sequestration, an elevation in oxidative stress, a decrease in antioxidants, gas exchange, expression levels of HIF-1α mRNA and lung function, in mice with endotoxemia treated with high-tidal-volume MV were substantially ameliorated in HIF-1α-deficient mice (*p* < 0.05; Figure 5 and Figure 6).

### 2.5. The Effects of Endotoxin-Stimulated MV-Induced Expression of Caspase-3 and Epithelial Apoptosis Are Partially Suppressed in HIF-1α Deficient Mice and Enoxaparin

In addition to its role in oxidative stress, caspase-3 is central to the intrinsic apoptotic pathway [20]. Capase-3 expression, terminal deoxynucleotidyl transferase-mediated dUTP-biotin nick end-labeling (TUNEL) staining and transmission electron microscopy (TEM) were performed to explore the functions of the caspase-3 pathway and the apoptosis of airway epithelial cells in endotoxin-associated VILI (Figure 7A). A substantial increase in caspase-3 expression and the appearance of TUNEL-positive apoptotic nuclei in the airway epithelia were observed in mice with endotoxemia treated with high-tidal-volume MV compared with the other MV treatment groups and the nonventilated control mice (Figure 7B,C). Epithelial apoptosis was confirmed by the characteristic nuclear condensation of bronchial epithelium in mice receiving MV and endotoxemia (Figure 7D). Specifically, there was a reduction in MV and endotoxin-exacerbated caspase-3 activity and apoptosis in the airway epithelia following the administration of enoxaparin. Levels of caspase-3 activity and apoptosis were also lower in the HIF-1α-deficient mice. Our results suggest that inhibiting the HIF-1α pathway leads to an inhibition of MV- and endotoxin-induced oxidative and inflammatory processes in the lungs (Figure 8).

## 3. Discussion

Sepsis is a severe infection associated with a profound systemic inflammatory response and life-threatening multiple organ failure caused by a dysregulated host response to infection [1,2,3]. Sepsis adversely affects the prognosis of critically ill patients admitted to ICUs. Novel therapeutic agents are needed to treat this serious clinical condition. Because acute respiratory failure often occurs with sepsis, patients require MV for life support. However, even MV with moderate tidal volumes can aggravate sepsis-induced lung injury by synergistically augmenting pulmonary cytokines, and this may play a central role in the evolution of ALI in patients with sepsis [21,22,23]. Thus, in this study, we employed our previously published animal model to simulate a clinical scenario to investigate the effects and molecular mechanisms of LMWH to attenuate VILI in endotoxemic mice [4,20]. We demonstrated that LMWH can (1) mitigate oxidative stress and improve antioxidant activity; (2) decrease inflammatory cytokines TNF-α, IL-6, MIP-2 and VEGF, as well as neutrophil sequestration; (3) ameliorate lung edema and microvascular leakage; (4) attenuate caspase-3 activity and epithelial apoptosis; (5) improve gross and microscopic epithelial pathology; and (6) restore the ultrastructural integrity, Penh physiological index and functional gas exchange in a mouse model of VILI with endotoxemia. Moreover, we investigated the role of HIF-1α in mediating the pathogenic mechanisms of VILI with endotoxemia.

HIF-1α is a critical regulator of cell metabolism with a high sensitivity to changes in intracellular oxygen concentration. Its expression level is determined by a balance between degradation and synthesis. In hypoxic conditions, HIF-1α accumulates, which functions as a signal to initiate the adaptive response to hypoxia, and then dimerizes with HIF-1β in the nucleus, leading to the transcription of various hypoxia-response genes, including VEGF and erythropoietin [9]. However, HIF-1α can also be stabilized and activated under normoxic conditions in a variety of situations, including contact with inflammatory cytokines (TNF-α, IL-6 and VEGF), exposure to ROS, cyclic mechanical stretch in vitro or by MV in vivo and infection [9,13,24,25,26]. Eckle et al. used in vitro and in vivo models to study the relationship between injurious mechanical ventilation and HIF-1α. They found that cyclic mechanical stretch induced repression of succinate dehydrogenase (SDH) and concomitant elevations of succinate levels. These products inhibited PHDs, leading to reduced polyubiquitination and proteasomal degradation and promoting HIF-1α stabilization, which is also confirmed in their in vivo study [13]. In this study, we demonstrated that MV, with or without LPS, activated HIF-1α gene expression; increased capillary permeability and lung edema; and caused inflammation, oxidative stress and apoptosis in endotoxemic mice. We also demonstrated that the deleterious effects were lessened in HIF-1α knockout mice. Mounting evidence suggests that HIF-1α can behave as a regulator of the progression and propagation of inflammatory processes [9,27]. HIF-1α also appears to have dichotomous roles in lung inflammatory diseases, as it can be injurious or protective depending on the physiological conditions as well as the nature and duration of the insult [9,13,28]. Haddad et al. indicate that a nonhypoxic pathway mediates cytokine-dependent regulation of HIF-1α translocation and activation in a ROS-sensitive mechanism [25]. Studies have implicated MV in the increase in HIF-1α expression in animal model lung tissue [13,24,28]. Thus, HIF-1α expression is not only responsive to hypoxia but also is evoked by proinflammatory cytokines, oxidants, cyclic stretch and LPS, according to our findings.

HIF-1α expression is acutely activated by LPS challenge in murine macrophages and human monocytes [29,30]. LPS-induced elevation of HIF-1α levels under normoxia may reflect a combination of accelerated HIF-1α transcription and decreased HIF-1α degradation mediated by PHD2 and PHD3 [11]. Combining MV and LPS challenge in our animal models modestly compromised the degree of arterial hypoxemia; however, HIF-1α expression was actively upregulated. This may be caused by MV-induced normoxic stabilization of alveolar epithelial HIF-1α, which is accomplished by inhibiting succinate dehydrogenase and sepsis-related inflammatory hypoxia [13,14]. Inflammation in a septic organism is frequently associated with tissue hypoxia due to several mechanisms, including decreased cardiac output, regional arteriolar vasoconstriction, decreased capillary flow, heterogeneity of microvascular perfusion, microthrombi due to endothelial dysfunction and leucosequestration [11,15]. In addition, proinflammatory cytokines inhibit oxygen utilization by mitochondria and produce a significant shift in oxygen demand beyond supply. Due to inflamed tissues becoming intensely hypoxic, a condition termed inflammation-associated hypoxia, cells metabolize ATP through glycolytic or anaerobic pathways [14]. During sepsis, HIF-1α, stimulated by LPS, activates the generation of inflammatory cytokines (including TNF-α, IL-1β and IL-6) and proapoptotic proteins, leading to inflammation and apoptosis [11,14,15]. Moreover, HIF-1α deletion in macrophages was protective against LPS-induced mortality [11,14,15]. In this study, HIF-1α knockout suppressed MV- and LPS-induced oxidants, TNF-α, MIP-2 and IL-6 and antioxidant activity was restored.

Thus, HIF-1α signaling is intensely involved in the vascular response to sepsis [26]. VEGF, a well-known HIF-1α target gene, has been demonstrated to be a critical mediator of sepsis-related morbidity and mortality [31]. VEGF response to several stimuli has been observed in vitro and in vivo and includes cellular stretch, LPS and HIF-1α [11,17,32]. In the early stage of ALI, increased production of alveolar VEGF may modulate vessels’ endothelial permeability and cause pulmonary edema by altering the state of the adherens junction complexes [28]. Ischemia and cytokines evoked VEGF gene expression. VEGF production appeared to increase DNA binding of HIF-1α to hypoxia-responsive elements in the VEGF gene promoter, resulting in microvascular leakage and leukocyte extravasation in lung and kidney injury [27,33]. Pharmacologic inhibition of HIF-1α activity significantly reduced the increased VEGF expression, angiogenesis, microvascular permeability and airway inflammation in lungs of H_2_O_2_-inhaled mice [34]. These findings are consistent with our results, which demonstrated the association of increased VEGF and microvascular permeability with lung edema and that neutrophil chemotaxis and sequestration caused by the combinatorial effects of MV and LPS were attenuated by the inhibition of HIF-1α. Taken together, our results accord with the previous findings that inhibition of HIF-1α activity could be a novel therapeutic target for the treatment of ALI and sepsis in animal studies and human shock states [11,15,35,36,37,38].

It has been demonstrated that LMWH prevents endotoxin-induced ALI and suppresses systemic inflammation in animal models of sepsis [39,40]. In the pathogenesis of sepsis, profound inflammation leads to activation of coagulation and coagulation also substantially affects inflammatory activity. Accumulating evidence has shown an extensive cross-talk between inflammation and coagulation [41]. Heparin was proven to exert protective effects of anticoagulation and anti-inflammation in animal models of VILI and sepsis [18,19]. A clinical meta-analysis researching the treatment effects of LMWH in septic patients reported that LMWH significantly improved prothrombin time, acute physiology, age, Chronic Health Evaluation II score and 28-day mortality and reduced platelet count compared with the standard treatment [7]. However, the precise molecular mechanisms of LMWH treatment for sepsis are still being determined. Li et al. reported that LMWH alleviated peritoneal fibrosis, possibly through suppression of HIF-1α, VEGF and TGF-β1 [42]. These results suggest that HIF-1α and VEGF, through the inhibition of inflammation-associated hypoxia, may play a central role in the beneficial effects of LMWH in the treatment of septic patients under MV. It has been determined that oxidative stress is initiated by mediators released from infiltrated neutrophils and activated lung macrophages that act on lung epithelial and endothelial cells. Moreover, oxidative stress activates the redox-sensitive transcription factors (NF-κB, activated protein-1), leading to a drastic production of proinflammatory cytokines and chemokines, which further exacerbates inflammation and oxidative stress [43]. These cascades lead to apoptosis of lung epithelial cells, which plays a critical role in the pathophysiology underlying MV- or sepsis-induced ALI [15,44]. In this study, we demonstrated that LMWH can inhibit oxidative stress, inflammation, apoptosis and increase antioxidant activity, eventually leading to a recovery of the pathological impairment through HIF-1α. Furthermore, LMWH increased oxygenation (PaO_2_/FiO_2_ ratio) and reduced the physiological index Penh, which is used to assess the changes of integrated adaption of breathing in pulmonary mechanics secondary to ALI [45].

In our study, partial inhibition of the increase of inflammatory cytokines and neutrophil sequestration in HIF-1α deficient mice suggested that HIF-1α signaling was only one of the many pathways contributing to the effects of LMWH on the inflammation-associated hypoxia [18,19,46,47]. Additionally, the nature of mouse respiratory mechanics is more compliant than human lungs and V_T_ up to 20 mL/kg is unlikely to induce substantial lung injury in models using healthy mice [48]. In our preclinical experimental study, we use animal study to simulate a clinical scenario, which is composed of multiple confounding factors associated with delivery of the drug to the affected tissues and toxicity towards essential parts of the organism. However, recent studies have demonstrated that HIF-1α is responsive to stimuli under normoxic conditions, such as thrombin [46] and LMWH is proven to inhibit plasma thrombin generation [47]. Further investigation using cell stretcher is required to determine whether the exploitation of LMWH therapy can rescue LPS-augmented VILI replicating the real clinical situation.

Overall, we demonstrated that LMWH inhibits oxidative stress, inflammation and apoptosis and leads to the substantial improvement of pathological, physiological and functional impairments by inhibiting HIF-1α signaling. Understanding the combinatorial effects of endotoxin and mechanical stretch on HIF-1α signaling may allow for the clarification of the molecular mechanisms in the pathogenesis of ALI related to sepsis and MV. Moreover, identification of LMWH that can reduce VILI in endotoxemic mice would provide a clinical therapeutic option for the management of ALI in septic patients requiring MV.

## 4. Materials and Methods

### 4.1. Ethics of Experimental Animals

Wild-type or HIF-1α-deficient C57BL/6 mice, weighing between 20 and 25 g, aged between 6 and 8 weeks, were obtained from Jackson Laboratories (catalog number 007227, Bar Harbor, ME, USA) and National Laboratory Animal Center (Taipei, Taiwan) [49]. The HIF-1αfl/fl, mouse line was bred to C57BL/6 mice carrying a CD4cre transgene. In the resulting offspring, a region encompassing exon 2 was excised in CD^4+^ cells. Genotyping on tail DNA was performed as previously described [50]. An amplification of ~250 bp by primers DP11 (5′-GCAGTTAAGAGCACTAGTTG) and DP12 (5′-GGAGCTATCTCTCTAGACC) indicated the presence of a floxed HIF-1α allele. An amplification of ~200 bp indicated a wild-type HIF-1α allele. PCR was used to genotype tail DNA for the presence of CD4cre (forward 5′-CGATGCAACGAGTGATGAGG, reverse 5′-CGCATAACCAGTGAAACAGC). CD4-Cre is the promoter element and HIF-1α is only knocked down in CD4+ cells [49]. The study was performed in strict accordance with the recommendations in the Guide for the Care and Use of Laboratory Animals of the National Institutes of Health (NIH). The protocol was approved by the Institutional Animal Care and Use Committee of Chang Gung Memorial Hospital (Permit number: 2017111001). All surgery was performed under xylazine and Zoletil anesthesia, and all efforts were made to minimize suffering.

### 4.2. Experimental Groups

Animals were randomly distributed into 7 groups in each experiment: group 1, nonventilated control wild-type mice with normal saline; group 2, nonventilated control wild-type mice with LPS; group 3, V_T_ 6 mL/kg wild-type mice with LPS; group 4, V_T_ 30 mL/kg wild-type mice with normal saline; group 5, V_T_ 30 mL/kg wild-type mice with LPS; group 6, V_T_ 30 mL/kg HIF-1α^−/−^ mice with LPS; group 7, V_T_ 30 mL/kg wild-type mice after enoxaparin (4 mg/kg) administration with LPS. In each group, three mice underwent TEM, and five mice underwent measurement for immunohistochemistry assay, inflammatory cytokines, protein carbonyl groups, SOD, TUNEL assay and western blots.

### 4.3. Ventilator Protocol

We used our established mouse model of VILI, as previously described [8,17]. Briefly, a 20-gauge angiocatheter was introduced into the tracheotomy orifice of mice and general anesthesia was maintained by regular intraperitoneal administration of Zoletil 50 (5 mg/kg) and xylazine (5 mg/kg) at the beginning of experiment and every 30 min. The mice were placed in a supine position on a heating blanket and then attached to a Harvard apparatus ventilator, model 55–7058 (Harvard Apparatus, Holliston, MA, USA), set to deliver either 6 mL/kg at a rate of 135 breaths per min or 30 mL/kg at a rate of 65 breaths per min for 5 h while breathing room air with zero end-expiratory pressure. At the end of the study period, heparinized blood was taken from the arterial line for analysis of arterial blood gas, and the mice were sacrificed. The nonventilated control mice were anesthetized and sacrificed immediately. 

### 4.4. Lipopolysaccharide Administration 

Mice will receive either 1 mg/kg of *Salmonella typhosa* lipopolysaccharide (Lot 81H4018; Sigma Chemical Co., St. Louis, MO, USA) or an equivalent volume of normal saline intravenously via the internal jugular vein as a control. After 1 h of spontaneous respiration to allow for development of a septic response, the mouse will be subjected to MV for 5 h [4].

### 4.5. Enoxaparin Administration

Enoxaparin (Sigma, St. Louis, MO, USA), 4 mg/kg, was given subcutaneously 30 min before MV, based on our previous in vivo study that showed 4 mg/kg enoxaparin inhibited blood coagulation and lung injury without significant bleeding tendency [8].

### 4.6. Whole Body Plethysmography

A noninvasive method (whole body plethysmography) was used to estimate Penh with or without MV [51]. The Penh reflects lung resistance changes in the waveform of the box pressure signal from both peak inspiration and expiration pressure. Each mouse was placed in a 3-in-diameter chambers (FinePoint^TM^ WBP, Data Sciences International, St. Paul, MN, USA) that was ventilated by bias airflow at 0.5 L/min/chamber. Mice were left to acclimatize inside the chambers for 5 min and then 3 min of respiratory responses were evaluated at baseline and after 5 h of MV.

### 4.7. Measurement of Inflammatory Cytokines

IL-6 with a lower detection limit of 1.8 pg/mL, MIP-2 (1 pg/mL), TNF-α (1.9 pg/mL), and VEGF (3 pg/mL) were detected in BAL fluid using immunoassay kits containing primary polyclonal anti-mouse antibodies that were cross-reactive with rat and mouse IL-6, MIP-2, TNF-α and VEGF (BioSource International, Camarillo, CA, USA). Each sample was run in duplicate according to the manufacturer’s instructions.

### 4.8. Real Time Polymerase Chain Reaction

For isolating total RNA, the lung tissues were homogenized in TRIzol reagents (Invitrogen Corporation, Carlsbad, CA, USA) according to the manufacturer’s instructions. Total RNA (1 μg) was reverse transcribed by using a GeneAmp PCR system 9600 (PerkinElmer, Life Sciences, Inc., Boston, MA, USA), as previously described [8]. The following primers were used for real-time PCR: HIF-1α, forward primer 5′-GCAGCAGGAATTGGAACATT-3′ and reverse primer 5′-GCATGCTAAATCGGAGGGTA-3′; and glyceraldehydes-phosphate dehydrogenase (GAPDH) as internal control, forward primer 5′-AATGCATCCTGCACCACCAA-3′ and reverse primer 5′-GTAGCCATATTCATTGTCATA-3′ (Integrated DNA Technologies, Inc., Coralville, IA, USA) [49,50]. All quantitative PCR reactions using SYBR Master Mix were performed on a CFX96 Touch Real-Time PCR Detection system (Bio-Rad Laboratories, Inc., Hercules, CA, USA). All PCR reactions were performed in duplicate and heated to 95 °C for 5 min followed by 40 cycles of denaturation at 95 °C for 10 s and annealing at 55 °C for 30 s. The relative gene expression was calculated using 2^−ΔΔ*C*T^ method and the standard curves (cycle threshold values versus template concentration) were prepared for each target gene and for the internal control GAPDH in each sample. The specific gene’s cycle threshold (*C*t) values were normalized to the GAPDH and compared with the nonventilated control group with LPS that was assigned a value of 1 to calculate the relative fold change in expression.

### 4.9. Immunoblot Analysis

The lungs were homogenized in 0.5 mL of lysis buffer, as previously described [8,16]. Crude cell lysates were matched for protein concentration, resolved on a 10% bis-acrylamide gel, and electrotransferred to Immobilon-P membranes (Millipore Corp., Bedford, MA, USA). For assay of HIF-1α, caspase-3 and GAPDH, western blot analyses were performed with respective antibodies (New England BioLabs, Beverly, MA, USA, Santa Cruz Biotechnology, Santa Cruz, CA, USA and Novus Biologicals, Littleton, CO, USA). Blots were developed by enhanced chemiluminescence (NEN Life Science Products, Boston, MA, USA).

### 4.10. Immunohistochemistry

The lungs were paraffin embedded, sliced at 4 μm, deparaffinized, antigen unmasked in 10 mM sodium citrate (pH 6.0), incubated with mouse HIF-1α (Novus Biologicals, Littleton, CO, USA) primary antibody (1:100; Santa Cruz Biotechnology, Santa Cruz, CA, USA), and biotinylated goat anti-rabbit secondary antibody (1:100) according to the manufacturer’s instruction for an immunohistochemical kit (Santa Cruz Biotechnology, Santa Cruz, CA, USA).

### 4.11. Histopathologic Grading of VILI

The lung tissues from control, nonventilated mice, mice exposed to high or low tidal volume ventilation for 5 h while breathing room air were removed en bloc and filled with 10% neutral buffered formalin (pH 6.8 to 7.2) at 30-cm H_2_O pressure via polyethylene tubing inserted into the trachea. The lungs were paraffin embedded, sliced at 4 μm, stained with hematoxylin and eosin and reviewed from 10 nonoverlapping fields by a single investigator blinded to the mouse genotype. Lung injury was scored using average of the following items: alveolar congestion, hemorrhage, infiltration of neutrophils into airspace or the vessel wall and thickness of the alveolar wall. A score of 0 represented normal lungs; 1, mild, <25% lung involvement; 2, moderate, 25% to 50% lung involvement; 3, severe, 50% to 75% lung involvement; and 4, very severe, >75% lung involvement [8].

### 4.12. Analysis of Data

The western blots were quantitated using a NIH image analyzer Image J 1.27z (National Institutes of Health, Bethesda, MD, USA) and presented as arbitrary units. Values were expressed as the mean ± SD from at least 5 separate experiments. The data of protein oxidation, SOD, histopathologic assay and oxygenation were analyzed using StatView 5.0 (Abascus Concepts, Cary, NC, USA; SAS Institute). All results of enhanced pause, real-time PCR and western blots were normalized to the nonventilated control wild-type mice with LPS. One-way ANOVA was used to assess the statistical significance of the differences, followed by multiple comparisons with a Scheffe’s test (for variable or same number per group), and a *p* value < 0.05 was considered statistically significant. Additional details, including oxidants, antioxidants, MPO, lung injury scores, TEM and TUNEL assay, and ventilator protocol were performed as previously described [8,17].

## Figures and Tables

**Figure 1 ijms-21-03097-f001:**
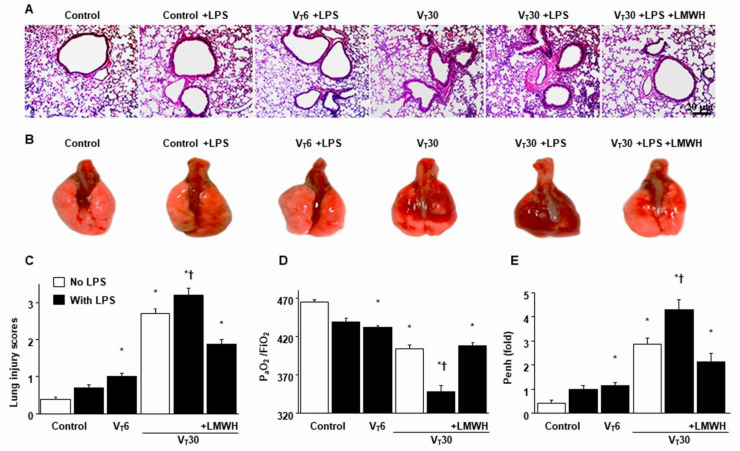
Suppression of endotoxin-augmented lung stretch-induced lung damage, hypoxemia and impaired respiratory function by enoxaparin (**A**) Histological examination (100×), (**B**) gross pathologic findings, (**C**) lung injury scores, (**D**) PaO_2_/FiO_2_ and (**E**) enhanced pause from the same animals were from the lungs of nonventilated control mice and those subjected to a tidal volume of 6 mL/kg (V_T_ 6) or 30 mL/kg (V_T_ 30) for 5 h with or without LPS administration (*n* = 5 per group). Enoxaparin, 4 mg/kg, was given subcutaneously 30 min before mechanical ventilation. Scale bars represent 20 μm. * *p* < 0.05 versus the nonventilated control mice with LPS; † *p* < 0.05 versus all other groups. FiO_2_ = fraction of inspired oxygen; LPS = lipopolysaccharide; LMWH = low-molecular-weight heparin; PaO_2_ = partial pressure of oxygen; Penh = enhanced pause.

**Figure 2 ijms-21-03097-f002:**
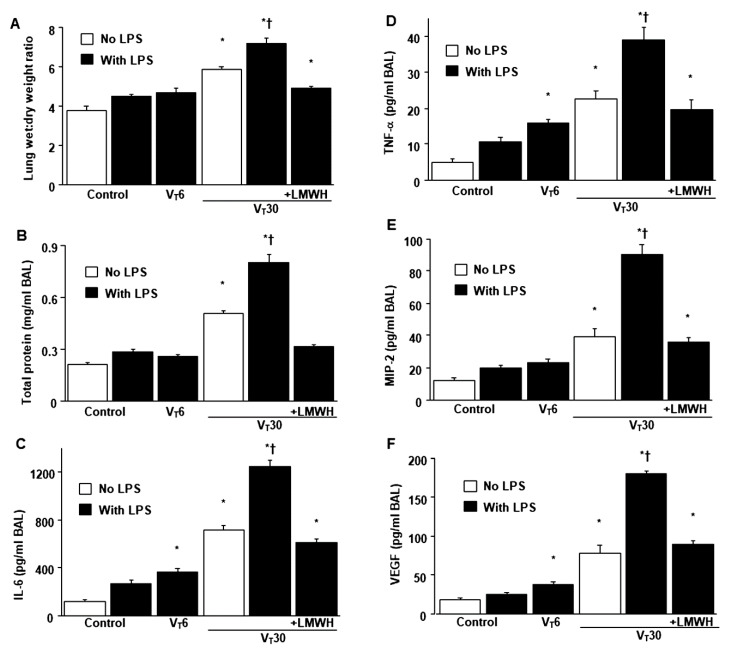
Inhibition of endotoxin-aggravated mechanical ventilation-enhanced lung injury by enoxaparin. (**A**) Lung wet-to-dry-weight ratio, (**B**) BAL fluid total protein, (**C**) IL-6, (**D**) TNF-α, (**E**) MIP-2 and (**F**) VEGF secretion in BAL fluid were from the lungs of nonventilated control mice and those subjected to a tidal volume of 6 or 30 mL/kg for 5 h with or without LPS administration (*n* = 5 per group). Enoxaparin, 4 mg/kg, was given subcutaneously 30 min before mechanical ventilation. * *p* < 0.05 versus the nonventilated control mice with LPS; † *p* < 0.05 versus all other groups. BAL = bronchoalveolar lavage; IL = interleukin; MIP-2 = macrophage inflammatory protein-2; TNF-α = tumor necrosis factor-α; VEGF = vascular endothelial growth factor.

**Figure 3 ijms-21-03097-f003:**
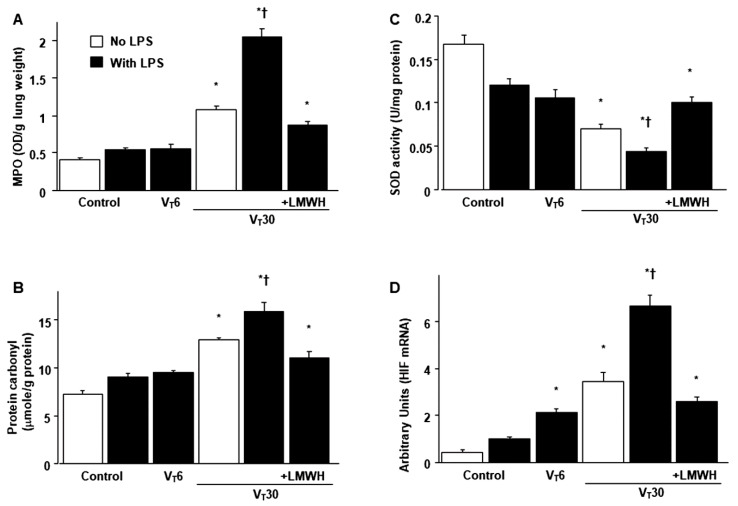
Reduction of endotoxin-exacerbated mechanical ventilation-mediated neutrophil influx, oxidative stress and HIF-1α mRNA expression by enoxaparin. (**A**) MPO activity, (**B**) protein carbonyl groups and (**C**) SOD from the same animals were from the lungs of nonventilated control mice and those subjected to a tidal volume of 6 or 30 mL/kg for 5 h with or without LPS administration (*n* = 5 per group). (**D**) Real-time polymerase chain reaction of HIF-1α mRNA expression was from the lungs of nonventilated control mice and those subjected to a tidal volume of 6 or 30 mL/kg for 5 h with or without LPS administration (*n* = 5 per group). Arbitrary units were expressed as the ratio of HIF-1α mRNA to GAPDH (*n* = 5 per group). Enoxaparin, 4 mg/kg, was given subcutaneously 30 min before mechanical ventilation. * *p* < 0.05 versus the nonventilated control mice with LPS treatment; † *p* < 0.05 versus all other groups. GAPDH = glyceraldehydes-phosphate dehydrogenase; MPO = myeloperoxidase; SOD = sodium dismutase.

**Figure 4 ijms-21-03097-f004:**
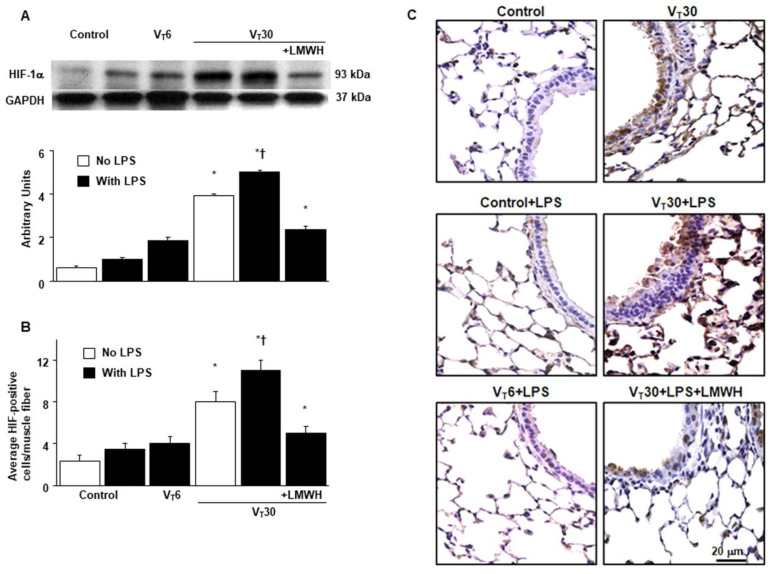
Abrogation of endotoxin-augmented mechanical ventilation-induced HIF-1α expression by enoxaparin. (**A**) Western blots from the same animals were conducted using antibodies that recognize HIF and GAPDH expression from the lungs of nonventilated control mice and mice ventilated at a tidal volume of 6 or 30 mL/kg for 5 h with or without LPS administration (*n* = 5 per group). Arbitrary units were expressed as the ratio of HIF-1α to GAPDH (*n* = 5 per group). (**B**,**C**) Representative micrographs (400×) with HIF-1α staining of paraffin lung sections and quantification were from nonventilated control mice and mice ventilated at a tidal volume of 6 or 30 mL/kg for 5 h with or without LPS administration (*n* = 5 per group). Enoxaparin, 4 mg/kg, was given subcutaneously 30 min before mechanical ventilation. Scale bars represent 20 μm. * *p* < 0.05 versus the nonventilated control mice with LPS; † *p* < 0.05 versus all other groups.

**Figure 5 ijms-21-03097-f005:**
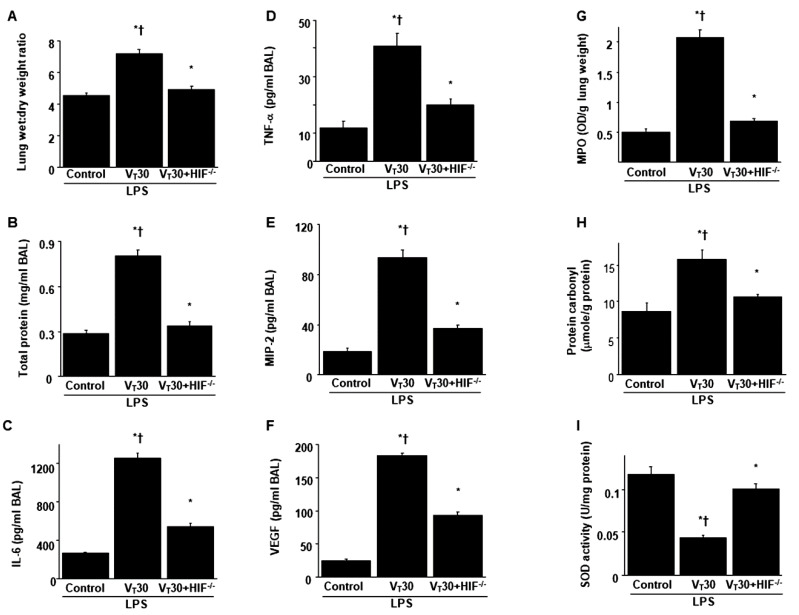
Inhibition of endotoxin-stimulated MV-induced lung injury in HIF-1α deficient mice. (**A**) Lung wet-to-dry-weight ratio, (**B**) BAL fluid total protein, (**C**) IL-6, (**D**) TNF-α, (**E**) MIP-2 and (**F**) VEGF secretion in BAL fluid, (**G**) MPO activity, (**H**) protein carbonyl groups and (**I**) SOD were from the lungs of nonventilated control mice and those subjected to a tidal volume of 30 mL/kg for 5 h with LPS administration (*n* = 5 per group). * *p* < 0.05 versus the nonventilated control mice with LPS; † *p* < 0.05 versus HIF-1α-deficient mice. HIF^−/−^ = hypoxia inducible factor-1α-deficient mice.

**Figure 6 ijms-21-03097-f006:**
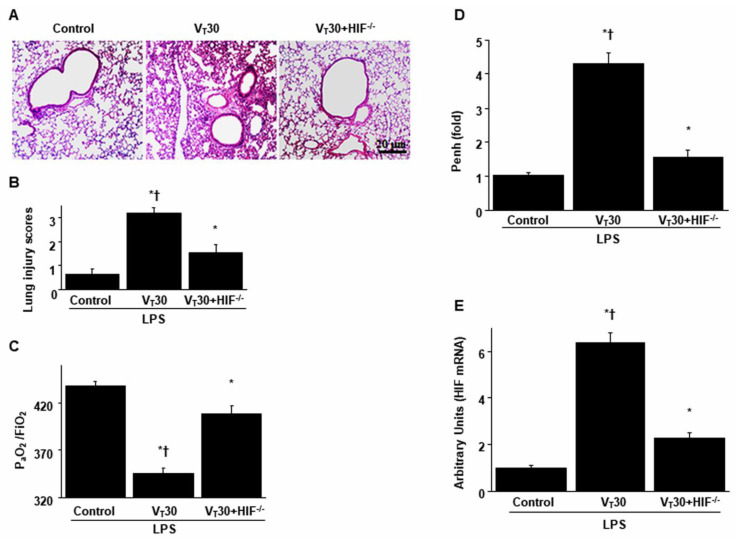
Reduction of endotoxin-aggravated mechanical ventilation-mediated lung damage, hypoxemia, impaired respiratory function and HIF-1α mRNA expression by HIF-1α homozygous knockout. (**A**) Histological examination (100×), (**B**) lung injury scores, (**C**) PaO_2_/FiO_2_, (**D**) enhanced pause and (**E**) HIF-1α mRNA expression were from the lungs of nonventilated control mice and those subjected to a tidal volume of 30 mL/kg for 5 h with or without LPS administration (*n* = 5 per group). (**D**) Real-time polymerase chain reaction of HIF-1α mRNA expression was from the lungs of nonventilated control mice and those subjected to tidal volume at 6 or 30 mL/kg for 5 h with LPS administration (*n* = 5 per group). Arbitrary units were expressed as the ratio of HIF-1α mRNA to GAPDH (*n* = 5 per group). Scale bars represent 20 μm. * *p* < 0.05 versus the nonventilated control mice with LPS treatment; † *p* < 0.05 versus HIF-1α-deficient mice.

**Figure 7 ijms-21-03097-f007:**
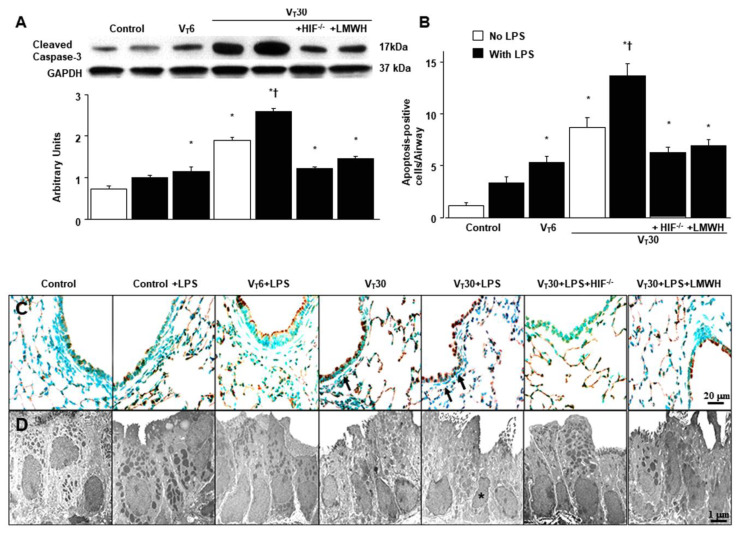
HIF-1α homozygous knockout and enoxaparin ameliorated endotoxin-augmented mechanical ventilation-induced caspase-3 expression and epithelial apoptosis. (**A**) Western blots from the same animals were conducted using antibodies that recognize caspase-3 and GAPDH expression from the lungs of nonventilated control mice and mice ventilated at a tidal volume of 6 or 30 mL/kg for 5 h with or without LPS administration (*n* = 5 per group). Arbitrary units were expressed as the ratio of cleaved caspase-3 to GAPDH (*n* = 5 per group). (**B**–**D**) Representative micrographs with TUNEL staining of paraffin lung sections and quantitation (**B**,**C**, 400×) (*n* = 5 per group) and transmission electron microscopic image (**D**, 10,000×) (n = 3 per group) were from the nonventilated control mice and those subjected to a tidal volume of 6 or 30 mL/kg for 5 h with or without LPS administration. A dark-brown diaminobenzidine signal indicated positive staining of apoptotic cells, whereas shades of blue–green to greenish tan signified nonreactive cells. Highly condensed and fragmented heterochromatin of epithelial cells indicates apoptosis. Apoptotic cells are identified by asterisks or arrows. Enoxaparin, 4 mg/kg, was given subcutaneously 30 min before mechanical ventilation. * *p* < 0.05 versus the nonventilated control mice with LPS; † *p* < 0.05 versus all other groups. Scale bars represent 20 μm (TUNEL) or 1 μm (TEM). TEM = transmission electron microscopy; TUNEL = terminal deoxynucleotidyl transferase-mediated dUTP-biotin nick end-labeling.

**Figure 8 ijms-21-03097-f008:**
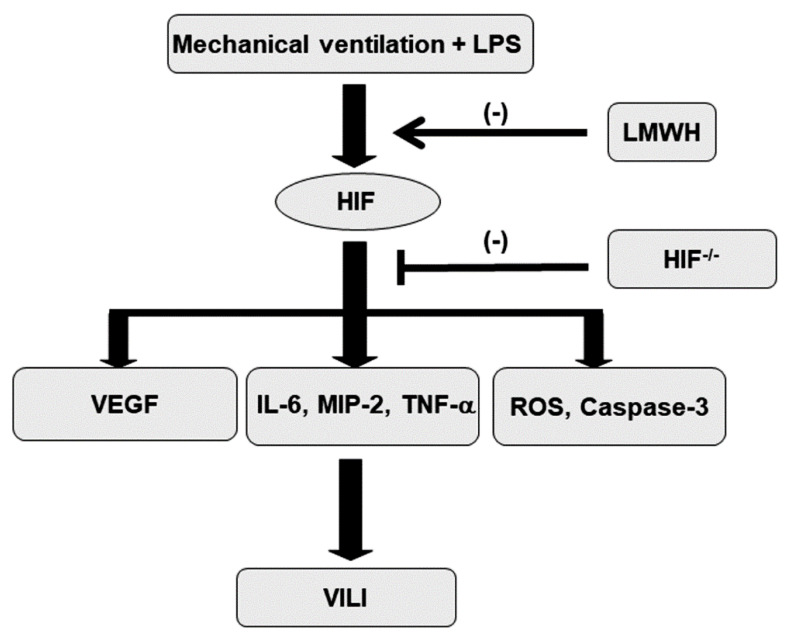
Schematic figure illustrating the signaling pathway activation with mechanical ventilation and endotoxemia. Endotoxin-enhanced augmentation of mechanical stretch-mediated cytokine production and lung injury were reduced by the administration of enoxaparin and with HIF-1α homozygous knockout. HIF = hypoxia-inducible factor; IL = interleukin; LMWH = low-molecular-weight heparin; LPS = lipopolysaccharide; MIP-2 = macrophage inflammatory protein-2; ROS = reactive oxygen species; TNF-α = tumor necrosis factor-α; VEGF = vascular endothelial growth factor; VILI = ventilator-induced lung injury.

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
