# Peer review of "Low-Molecular-Weight Heparin Reduces Ventilation-Induced Lung Injury through Hypoxia Inducible Factor-1α in a Murine Endotoxemia Model"

_ijms, 2020, doi:10.3390/ijms21093097_

Round 1
Reviewer 1 Report
Authors describe that Low-Molecular-Weight Heparin (LMWH) suppresses hypoxia inducible factor-1alfa (HIF-1alfa), and it also ameliorates ventilation-induced lung injury. They show that various inflammatory cytokines are suppressed by LMWH administration. Their results suggest the role of HIF-1alfa in ventilator associated lung injury. Their work also suggests the therapeutic possibility of LMWH. I have just two questions bellow.
How does mechanical ventilation lead to the production of HIF-1alfa?
How does HIF-1alfa induce various inflammatory cytokines?
Reviewer 2 Report
Li-Fu Li and coworkers report an involvement of Hif1A in the pathogenesis of ventilator-induced lung injury (VILI) during endotoxemia. In a murine model of VILI with LPS treatment, they show how high tidal mechanical ventilation (MV) leads to increased lung injury, production of free radicals, inflammatory cytokines and increased levels of HIF-1a. Additionally, the observed effects in LPS treated and high tidal ventilated animals, including increased HIF-1a expression, are shown to be reversible by enoxaparin treatment. As they observe a reduced lung injury in HIF-1a deficient animals, similar to the reduction upon HIF-1a treatment, they suggest that enoxaparin inhibits the HIF-1apathway, which reduces endotoxin-augmented MV-induced ALI.
In my opinion, many of the claims are not well substantiated or are based on over-interpretation of data. In addition, the novelty of the involvement of HIF-1a as well the protective effect of heparins in VILI development and/or endotoxemia is questionable. The mechanistic conclusion, that Heparin directly acts on HIF-1a and thereby reduces lung injury remains unclear. If the authors are convinced of the novelty of their findings, I would suggest to focus on one story - either delining the role of HIF-1a or describing the protective effect of enoxaparin. This would reduce the amount of figures and enhance the clarity of their message.
Major concerns:
- It remains unclear, why the authors investigate HIF-1a in the context of VILI development in the first place. As the authors discuss, the effects observed during to VILI and sepsis on Hif1A expression are already known for hypoxemia and sepsis models, so that novelty of these results is questionable.
Overall inflammatory markers, cell recruitment and ROS production demonstrate a similar dynamic as HIF-1a in their model. It remains unclear why increased inflammation in the combined model of VILI+LPS should be particularly due to HIF-1a and not just coinciding effects.
- The generation of HIF-1a KO mice in the methods is not clear. Is CD4-Cre the promoter element? Is HIF-1a only knocked down in CD4+ cells?
- The additional effect of LPS stimulation on VILI is broadly known in the field. The authors lack the explanation why LPS and sham stimulation upon MV are just applied to non ventilated/ high tidal groups and not to low tidal groups. Non ventilated animals should serve as baseline, but the correct control groups for experiments with high tidal ventilation are low tidal ventilated animals. The applied narcosis and time of MV have to be matching in the control. Do low tidal ventilated animals also show an increased inflammation up on LPS stimulation compared to control? Is inflammation in the low tidal group also reversible upon enoxaparin treatment? The use of low tidal controls could greatly empower the clinical/scientific relevance of their results. For the conclusions drawn in their discussion, low tidal MV controls (+/-LPS, +/-treatment) should be mandatory.
- To test the protective effect of enoxaparin treatment, the authors focus on HVT groups. As mentioned before, it remains unclear, what effect enoxaparin would have in low tidal ventilated animals (supplementary figures are needed)?
- The authors’ claim, that enoxaparin is decreasing VILI development through HIF-1a, but this is not backed by in vitro data. To strengthen the mechanistic conclusion of this paper in vitro experiments for of cytokine release (epithelial / endothelial cells) and ROS production e.g. assays are needed.
- Is Enoxaparin inhibiting thrombin activation and thereby HIF-1 release during VILI? Or is Enoxaparin protective by blocking sepsis-induced coagulation? There are several studies published in mice and other small animal models of VILI/Sepsis that show a protective effect of heparin treatment (Tuinman et al., Crit. Care 2012; Camprubí-Rimblas et al. ,Ann. Transl. Med. 2018).
Minor concerns:
- The abstract does not reflect well the results and conclusions drawn in the paper.
- The result section needs to be more specific and describe the single figures and observed significances in detail.
- The title “The effects of MV on enoxaparin-exacerbated HIF-1a expression by enoxaparin.” The reviewer suggests, that endotoxin is mend in the first place?
- The title “Suppression of the Effects of MV on Endotoxin-Augmented VILI by Enoxaparin” and following are confusing and should be changed. There is only data for high tidal MV and enoxaparin treatments shown, low tidal controls are lacking. Furthermore it is not clear, if it is the effect of MV on endotoxin-augmented VILI, or rather the effect of endotoxin stimulation on VILI, that is partially inversed by enoxaparin.
- The title “Inhibition of the Effects of MV on Endotoxin-Enhanced Lung Injuries, Hypoxemia, and Impaired Respiratory Function by HIF-1 Homozygous Knockout” Are different lung injuries examined or is the lung injury quantified by different parameters? “Inhibition of the effect… by HIF-1a homozygous knockout” – Maybe rather “a decreased lung injury in HIF-1a deficient animals”? Overall, spelling and grammar needs major revisions.
- Figure 1B: photos of whole lungs are not sharp and not representative. Further histologic pictures and single histologic scorings for hemorrhage, intra-alveolar cells and PAS staining for hyaline membranes would be preferable.
- Figure 4 C: The graphic representation is confusing. The image labeling is difficult to decipher against the background. In the first row LPS stimulation is on the right side. In the second row LPS stimulation is on the left side, but the right image is not labeled. Is it a second picture of LVT+LPS group to have equal numbers of images in each row? In the third row the left and right image are with LPS stimulation.
- Figure 5: “Inhibition of endotoxin-aggravated lung stretch-enhanced lung injury by HIF-1a homozygous knockout.” The word “lung stretch-enhanced” is confusing and wrong. Stretch has never been quantified in this model of MV. Stretch is rather adapted for in vitro cell stretch experiments.
- Generally exemplary western blots are badly represented and seem cut together from different blots. Photos of whole gels including the respective controls would increase the credibility of the data.
- Statistics: Are statistical analyses for western blots done with replica samples (i.e. from the same experiment) or from truly independent experiments? This needs to be clarified in each figure legend. Are single results (e.g. P/F ratio, oxygenation, BAL protein etc.) for each animal experiment (n=5) obtained from the same animals? Statistical tests and error bars used have to be reported in each figure. Specify which Anova has been used: 1way-anova/2way-anova? Sheffe’s multi comparison test is not adapted, use Turkey (preferable) or Sidak.
